# A New Device for Measuring Trunk Diameter Variations Using Magnetic Amorphous Wires

**DOI:** 10.3390/s25144449

**Published:** 2025-07-17

**Authors:** Cristian Fosalau

**Affiliations:** Faculty of Electrical Engineering, “Gheorghe Asachi” Technical University of Iasi, 700050 Iasi, Romania; cristian-ioan.fosalau@academic.tuiasi.ro

**Keywords:** dendrometer, trunk diameter measurement, magnetic amorphous wires, stress impedance effect

## Abstract

Measuring the small tree trunk variations during the day–night cycle, seasonal cycles, as well as those caused by the plant’s growth and health regime is a very important action in horticulture or forestry because by analyzing the collected data, assessments can be made on the health of the trees, but also on the climatic conditions and changes in a certain region. This can be performed with devices called dendrometers. This paper presents a new type of approach to these measurement types in which the trunk volume changes are highly sensitively converted into the axial stress on sensitive elements made of magnetic materials in wire form in which the giant stress impedance effect occurs. Finally, by electronic processing of the signals provided by the sensitive elements, digital words with a decimal value proportional to the diameter variations are obtained. This paper presents the operating principle, the constructive details and the experimental results obtained by testing the device in the laboratory and in-field. The proposed dendrometer, compared to those available commercially, has the advantage of good resolution and sensitivity, good immunity to temperature variations, the possibility of transmitting the result remotely, robustness and low price. Some metrological parameters obtained from the experimental testing are the following: resolution 1.6 µm, linearity 1.4%, measurement range 0 to 5 mm, temperature coefficient 0.012%/°C.

## 1. Introduction

Agriculture is one of the most important human activities because it is the main source of food for people. Traditional methods of farming no longer meet the needs of the ever-growing population. Therefore, new methods are being sought using the advantages offered by current science and technologies to increase agricultural productivity, while maintaining quality as much as possible. There are numerous possibilities for increasing productivity in agriculture, including the use of selected seeds, fertilizers and pesticides, the adoption of sustainable agricultural practices, digital farm management, production diversification, but also the use of modern technologies such as complex machinery and equipment, drones and, last but not least, Internet of Things (IoT) technologies for monitoring specific parameters [1,2,3]. Within an IoT system, the primary element is the sensor. It provides information relating to the measured quantity in the form of analog or digital signals, which are received and transmitted by various methods, usually wirelessly, to collection servers that further process the information and present it in an accessible form to the user.

In general, the dimensions of a plant, starting from the petiole and leaf, to the branches, trunk and root, change due to variations in the water potential circulating through the plant’s conducting vessels, a potential that is provided by the water content in the plant [4].

Dendrometry is a science used in forestry and particularly in horticulture, which measures various dimensions of plants and trees, such as trunk and branch diameter, crown diameter, wood mass volume, tree shape and age, height, variations in these dimensions over time, fruit size, sap flow through woody vessels, etc. Knowledge of these dimensions leads to a better understanding of the health of trees, their growth performance and a better exploitation of the benefits brought by plantations and forests. Monitoring the relative variations in trunk volume over certain periods of time can provide important information regarding water stress [5], nutrient deficiency [6], pest and insect occurrence [7], soil compaction degree [8], etc. For example, determining the degree of humidity and water requirements for the plant, in conjunction with the meteorological and environmental factors measured with IoT systems, allows the optimal irrigation of plants to be programmed so that water is not supplied in excess, causing waste, but neither is the water requirement not sufficiently met, leading to plant suffering [9]. In [10], a study was presented on how and to what extent variations in trunk diameter can provide information on the health status and in particular, the water deficit of an olive plantation using dendrometric measurements. It was established that this information exists and can be useful, but it must be corroborated with other influencing factors and other measured quantities such as a series of meteorological parameters, soil water content, leaf water potential, carbon dioxide assimilation rate, stomatal conductance, etc.

In particular, a dendrometer is a device that can measure the diameter of a tree trunk or, more precisely, the very small variations in these dimensions owed to the aforementioned causes. A few decades ago, dendrometers were of the mechanical type in the form of calipers with which the trunk diameter was manually measured at certain intervals. The method was both imprecise and lacked some essential information that could have been obtained by continuous monitoring such as certain key moments of diameter fluctuations from which valuable conclusions can be drawn regarding the health state and needs of the plant. The method is still used today, with the specification that the instruments are electronic and are able to measure the diameters with good precision [11].

An ingenious method for measuring the diameter of cylindrical objects with application in measurement using forest calipers is described in [12]. The principle is based on laser telemetry under structured lighting, where a laser plane creates a line on the object under test, which is observed by a charge-coupled device camera. The tree radius is calculated by the novel “three-tangents method”. The reported accuracy is relatively good (1%), but it is affected by the cylindricity of the tree.

With the development of technologies, especially digital ones and especially continuous distributive measurement technologies through IoT networks, dendrometer-type instruments have become more complex and more efficient. With such instruments, which can automatically and continuously measure trunk diameter, it is possible to monitor and capture important critical moments in the life of the plant such as water shortage, the appearance of pests or the need for fertilizers. In general, the dendrometers currently on the market are divided into two categories, according to the operating principle:Non-contact dendrometers–these devices measure small displacements in the tree trunk by optical interferometric methods [13] or using high-resolution video cameras [14].Contact dendrometers–variations in the trunk diameter or circumference are transformed into linear or angular displacements usually measured with an inductive, capacitive or resistive displacement transducer.

Referring to the last category, in [15], a method for estimating the water potential in a plant by measuring the diameter of a leaf petiole with a dendrometer built with the aid of a linear potentiometer displacement transducer calibrated on the basis of an optical dendrometer is described. The proposed system turns out to be very cheap and relatively accurate but suffers from low reliability due to moving and frictional elements. A dendrometer whose operating principle is based on the conversion of circumference variations into linear displacement measured using a film potentiometer is described in [16]. Another possibility, presented in [17], refers to the calculation of the trunk diameter from the electronic measurement of the opening angle of a caliper and other geometric dimensions. The method is very imprecise but cheap. The paper [18] presents a method for measuring the diameter using amorphous magnetic materials similar to those used in the transducer developed in this paper. In this case, the wire is wrapped around the trunk and the impedance variation is measured by the giant magnetoimpedance (GMI), effect, which occurs due to the stress applied to the wire by the increase in volume of the trunk. The method requires the use of a large amount of expensive material. Finally, we can also mention here an interesting possibility of indirect measurement of the tree trunk by evaluating its age using tree-ring image analysis using machine learning algorithms, as described in [19]. This is, however, a very imprecise invasive method, devoted more to foresters to study the events that occurred during large periods of time (tens and hundreds of years).

This paper presents the operating principle, construction and performance of a dendrometer designed to measure small variations in the diameter of tree trunks, which occur in the succession of seasons, the day–night cycle, as a result of variations in environmental conditions or the health of the tree. The operation of this device is based on the properties of special materials called magnetic amorphous wires (MAWs). In the following sections, the giant stress impedance (GSI) effect on which the operation of the sensitive element (SE) of the device is based, the construction of the mechanical and electrical components, as well as the experimental results in terms of the characteristics and performances of the device will be presented.

It should be noted here that the proposed device is not intended for the absolute measurement of the tree trunk, but for its relative variations with respect to a spatial and temporal reference point, usually set and calibrated at the mounting of the dendrometer on the tree, the variations caused by the parameters mentioned above.

The device proposed and described in this paper exhibits superior qualities to existing dendrometers due to its very good sensitivity, robustness and low price.

## 2. Operation Principle of the Dendrometer

The operation principle of the proposed dendrometer is depicted in Figure 1. It consists in transforming the variation in the trunk diameter into a linear movement, which will be further converted into an electrical signal by the SE through the GSI effect.

The flexible strip 2, of constant length L, made of stainless steel, is wrapped around the trunk 1. Since temperature is one of the most important environmental factors affecting the accuracy of measurement with this transducer, the strip was chosen to be made of stainless steel because this material has a relatively low thermal expansion coefficient, good elasticity, mechanical strength and low price, and is also protected against corrosion.

With a good approximation, we can consider the section through the trunk of the tree to be oval in shape, an ellipse with the major and minor axes a and b.

The increase in volume of the trunk produces a variation d (the increase is assumed to be uniform along the two axes) and hence of the perimeter, which leads to the displacement of the mobile point from position M_1_ at a distance X_1_ from the trunk surface to position M_2_, at the distance X_2_ from the trunk surface. Obtaining an analytical relationship of dependence between dx and d is difficult, but this operating principle was simulated in Autodesk’s Fusion 360 v.2.0 software, obtaining an approximately linear dependence:(1)$$dx=F\left(a,b,X_{1}\right)x=m_{d}d$$

The above equations works under the simplifying conditions d << a,b and a/b << 1.15. An example of such a dependence is given in Figure 2, traced for X_1_ = 20 mm, a = 155 mm, b = 140 mm.

Following this study carried out with Fusion 360, the following conclusions were drawn:the slope (sensitivity) and the linearity of the characteristic (1) depend on the a/b ratio, the X_1_/a ratio and the mounting position of the device;the maximum linearity is obtained when the tree section is circular; yet, if the tree has eccentricity, the sensitivity increases with the increase in the a/b ratio, but the linearity is reduced to the same extent. It is therefore necessary to measure first the two axes, and the device should be mounted along the major axis, when the contact angle of the strip with the trunk leads to the best contact of the strip to the tree bark;increasing the initial distance X_1_ of the moving part from the trunk surface, relative to the major axis, decreases the sensitivity, but improves the linearity of the characteristic.

In the example above, it was assumed that the tree trunk section was elliptical. In reality, nature is extremely diverse, and things are much more complex. First of all, the section may be of highly irregular shape, in which case this dependence is unpredictable, and the actual calibration of the device can only be performed in the sensor part, as will be shown in Section 3.2. Also, the growth in diameter can be anisotropic, i.e., it can differ along different radial directions. On the other hand, as previously mentioned, this device is not intended for the precise measurement of absolute variations in the diameter of trees, but for relative determinations through which, starting from an initial spatial and temporal point of reference, the state of the plant can be assessed over a certain period of time. With this regard, even the usual commercial dendrometers undergo similar drawbacks.

The variations in the trunk diameter d have, therefore, been converted into a linear displacement dx, which needs be further measured with a displacement transducer. In the construction of the existing dendrometers on the market, this displacement is measured, for example, using an inductive displacement transducer, achieving a resolution of 5 μm at the most [14].

Based on this principle, in the following section, a device will be described in which the conversion of the movement dx into an electrical signal is performed by means of a force transducer built with magnetic amorphous wires.

## 3. The Sensitive Element

The sensitive element of the device is a wire with a special structure and composition, sensitive to mechanical stresses, in which case it changes its electrical impedance. This wire is used in the present application as an axial force sensor, based on which the value of the displacement of the mobile component *dx* of the device is measured. The main advantage of this material is its very good sensitivity to variations in the axial force applied to it, which gives a high sensitivity to the entire transducer.

### 3.1. About Magnetic Amorphous Wires (MAWs)

MAWs are materials with the structure M_x_Si_y_B_z_ where M is a metal or a combination of metals such as Fe, Co, Ni, Cr, Mn, Cu and Nb in various proportions, manufactured in the form of wires or ribbons, having an amorphous structure. They are produced by a process called the “in-rotating-water-quenching-method” in which the alloy components are introduced in the desired proportions into an induction furnace and melted in an argon atmosphere, to avoid oxidation [20,21,22]. Through a nozzle located in the lower part of the furnace, the molten metal is discharged under pressure into a layer of cold water located on the inner wall of a drum rotating at high speed. The jet of molten metal solidifies rapidly upon contact with the cold water, the material remaining in an amorphous state, resulting in a continuous wire that is deposited in the form of a coil on the inner wall of the drum. By optimally adjusting technological parameters such as alloy melting temperature, nozzle inclination angle and drum rotation speed, a continuous, homogeneous wire with a circular cross-section is obtained.

This process can result in two types of wires: (i) MAWs with high magnetostriction (e.g., Fe_77.5_Si_7.5_B_15_), where the magnetostrictive coefficient is λs > 30 ppm and (ii) MAWs with low magnetostriction and high permeability (e.g., Co_94_Fe_6_)_72.5_Si_12.5_B_15_), where λs < 0.1 ppm and the relative permeability μ > 40,000 [21,22].

The internal structure of the magnetic domains of such a wire consists of a longitudinal core covering about 70% of the wire volume, which forms a single magnetic domain with axial magnetization, and a shell in which the magnetic domains are arranged radially or circularly with respect to the wire axis. Depending on the domain structure, composition and type of the MAW, a series of special effects and phenomena occur, which can be exploited in the construction of a large number of types of sensors such as magnetic field, force, displacement, current, deformation, pressure sensors, etc. One of these effects is the giant magnetoimpedance effect (GMI), from which the giant stress impedance (GSI) effect is derived.

GMI is due to the internal structure of circular domains in the shell and to the internal mechanical stresses that arise during the rapid solidification process of the wire and occurs in MAWs with a low magnetostriction. Essentially, the GMI effect consists of the modification of the impedance of a MAW when it is supplied with an alternating electric current of a certain frequency and is subjected to an axial magnetic field [23]. The impedance of the wire, Z, depends on the frequency f of the ac current that passes through the wire, on the intensity of the axial magnetic field H, but also on the axial and radial mechanical stresses to which the wire is subjected. Under conditions in which the frequency and the magnetic field are kept constant in a certain configuration, the impedance of the wire depends on the mechanical stresses applied to the wire, in which case the GMI effect is called the GSI effect. The two effects, GMI and GSI, are in fact a consequence of the skin effect that occurs in magnetic materials traversed by electric current and are due to the variation in the permeability of the material at the surface, in principle due to the interaction between the magnetization of the highly permeable magnetic domains of the shell and the mechanical stresses in the wire. The GSI effect is due to the modification by the external axial stress action of the internal configuration of the residual stresses of the MAW, usually in arbitrary directions, produced during the rapid quenching and solidification of the wire, which leads to its non-crystalline but amorphous structure.

The Z impedance of the MAWs is composed of the two real and imaginary components, Z_r_ and Z_im_:(2)$$\underline{Z}=Z_{r}+jZ_{im}$$
For low values of the frequency (f < 100 kHz), Z_r_ is given mainly by the direct current resistance of the wire, R_dc_, whereas Z_im_ depends, through the inductance L of the wire, on the length l of the wire, on the circular permeability μ_Φ_, which in turn depends on the axial component of the magnetic field, H_x_, and on the axial stresses σ and circular (torsional) stresses ξ applied to the wire [24,25] according to the following:(3)$$\underline{Z}=R_{cc}+j\omega L,\;\;\;\;\;\;\;L=\frac{l\mu_{\phi }\left(H_{x},\;\xi ,\;\sigma \right)}{8\pi }$$
In turn, for higher values of the frequency, both components depend on the skin effect by the penetration depth, δ, given by [24,25]:(4)$$\delta =\sqrt{\frac{2\rho }{\omega \mu_{\phi }\left(H_{x},\;\xi ,\;\sigma \right)}}$$
where ρ is the material’s resistivity and ω is the current pulsation. In these conditions, Z becomes the following:(5)$$\underline{Z}=\frac{r}{2\sqrt{2\rho }}R_{cc}(1+j)\sqrt{\omega \mu_{\phi }\left(H_{x},\;\xi ,\;\sigma \right)}$$
One can observe from (5) the dependence of Z on the stress σ, from which the stressimpedance name of this effect originates.

### 3.2. The Sensitive Element Characteristics

As stated above, for the construction of the dendrometer, the GSI effect was employed, that is, the change in the impedance of a MAW when it is subjected to an axial stress, σ. An important stage in the design of the device was the identification and optimization of the SE in terms of its sensitivity and linearity in order to obtain the best final results for the device.

The SE or the sensor is built from a MAW with the composition (Co_94_Fe_6_)_72.5_Si_12.5_B_15_ of diameter 120 μm ± 4 μm, produced by the National Institute for Research and Development for Technical Physics Iaşi, Romania. When optimizing the SE, the parameters taken into account were wire length, current frequency and intensity. The experimental characteristics Z(σ) and the relative variation in the impedance, ΔZ_r_, were experimentally plotted, for the employed MAW. ΔZ_r_ denotes the GSI effect sensitivity and is expressed as follows:(6)$$\Delta Z_{r}\left(\sigma \right)=\left|\frac{\left|Z\left(\sigma \right)|-{|Z}_{0}\right|}{{|Z}_{0}|}\right|100\;\;\;\;[\%]$$
where Z(σ) is the MAW impedance under stress σ, whereas Z_0_ is its impedance in relaxed state. In the above relation and in all that follows, we refer only to the impedance modulus, noting Z ≡ |Z|.

The experiment was performed for 3 different wire lengths: l = 10 mm, 20 mm and 30 mm and at various values of the ac current frequency, f = 100 kHz, 1 MHz, 2 MHz, 3 MHz and 4 MHz. A schematic of the experimental setup with which the characteristics were plotted is given in Figure 3. It is composed of a platform on which a fine pitch screw stresses the MAWs by means of a calibrated spring whose elastic constant is k = 296 N/m, that transforms the displacement of a mobile piece into the force F that acts axially upon the MAWs, in which the stress σ occurs. In turn, σ produces an impedance variation through the GSI effect, which is measured in terms of its module and phase using an AGILENT 4285A automatic bridge working in polar coordinates. All measurements were performed automatically using a virtual instrument built in the LabVIEW programming environment that drives the whole acquisition process, the only manual action being the actuation of the wire pre-stressing fine-pitch screw at the measurement points. The range of stress variation was between 0 and 450 MPa which corresponds to 12 mm displacement of the mobile piece.

In Figure 4, some examples of such identification characteristics are given, representing the Z(σ) variation for a constant length, intensity and variable frequency, the variation ΔZ_r_ (σ) for constant frequency, intensity and various lengths and the final SE characteristic with optimized parameters, which was further used in the design of the dendrometer.

As a result of this study, the following were found:The current intensity influences the shape of the characteristics to a very small extent. Therefore, the current of approximately 1 mA was considered optimal, a compromise between consumption, signal-to-noise ratio and the amount of heat released in the SE. The value of 1 mA is not critical.The impedance of the wire increases with its length and frequency.Shorter wires have more linear characteristics but are less sensitive to the GSI effect than longer wires.

Increasing the frequency improves the linearity of the characteristics, the sensitivity remains approximately constant for a range of σ between 0 and 300 MPa, of approximately 30%. As noticed from the above observations, it is necessary to make a trade-off between the sensitivity of the GSI effect measured with (6), which is actually the SE sensitivity, and its linearity. As a result of this study, the following optimal values for the SE were found: composition (Co_94_Fe_6_)_72.5_Si_12.5_B_15_, Φ = 120 ± 4 μm, l ≈ 20 mm, I ≈ 1 mA, f ≈ 1 MHz. For these parameters, the characteristic in Figure 4d was drawn. This characteristic is not actually linear, but by restricting the range of variation in the axial stress, a working range centered on a bias point (bp) M(σ_bp_,Z_bp_) with a linearity below 1% can be set. The point M is obtained by pre-stressing the wire with the constant stress σ_bp_. On this linear area, the following equation is written:(7)$$Z_{lin}(\sigma )=Z_{bp}+S_{bp}(\sigma -\sigma_{bp})$$
where S_bp_ is the sensitivity of SE with respect to the stress, given by the following:(8)$$S_{bp}={\frac{\partial Z}{\partial \sigma }|}_{\sigma =\sigma_{bp}}$$

At this point, S_bp_ = 0.0125 Ω/MPa.

For any other point on the linear characteristic, under the action of stress σ, an impedance variation $\Delta Z_{\sigma }$ occurs, which is given by the following:(9)$$\Delta Z_{\sigma }=Z_{lin}\left(\sigma \right)-Z_{bp}=S_{bp}(\sigma -\sigma_{bp})$$

## 4. The Dendrometer Construction

The dendrometer consists of a hardware structure made of a mechanical part whose components were 3D printed, in which the SEs are mounted in an electrical bridge structure and the electronic signal conditioning circuit dedicated to processing the signal provided by the SE bridge and transforming it into an electrical voltage proportional to the diameter variations in the trunk to which the device is attached.

### 4.1. Mechanical Structure

A simplified diagram of the main structure of the dendrometer is given in Figure 5.

According to Figure 5, the whole device is attached to the trunk 1 of the tree by using the strip 2 that acts upon the shoulder 4 which slides in relation to the fixed casing 3. The movement of the shoulder 4 in the radial direction towards the trunk is due to the change in the trunk diameter, as described in Section 2, Figure 1. The movable piece 5 is attached to the shoulder 4, and the piece 7 is attached to the fixed casing 3. Thus, a relative movement of the pieces 5 and 7 occurs, between which the springs 6 are found, which convert this movement into the force F according to the relationship:(10)F=kdx
where dx is the displacement defined in Figure 1 and k is the spring constant. Parts 5 and 7 are attached to parts 9 and 10 that act on the MAW 11 which is fixed by means of screws A, B, C and D so that sections BC and AD are subjected to an axial tensile stress σ produced by the force F, given by the following:(11)$$\sigma =\frac{F}{S_{MAW}}$$
where *S_MAW_* is the section of the wire. For the wires employed in this approach, S_MAW_ = 3.6 × 10^−10^ m^2^. We will call these wires active elements (AEs). In turn, the sections AB and CD are not subject to axial stresses (passive elements, PEs), but under the influence of the same external factors (such as temperature and external magnetic fields) as AEs. The idea of using four elements connected in a complete bridge, two active and two passive, has two advantages: (i) it increases the sensitivity of the device and (ii) it allows for the compensation in the temperature and other influential factors over the SE. Screw 8 is provided for pre-stressing the MAW to bring it into the linear working area, viz. to the bias point M. A Wheatstone bridge configuration is thus obtained as detailed in Figure 6a, whose electrical diagram is presented in Figure 6b.

### 4.2. Mathematical Approach

According to Figure 6, the active elements AE_1_ and AE_2_, characterized by impedances Z_a1_ and Z_a2_, are positioned on opposite sides of the bridge, as are the passive elements PE_1_ and PE_2_, characterized by impedances Z_p1_ and Z_p2_.

The bridge is powered with alternating current I (here only the amplitude of this current is of interest) between points A and C of the bridge, and the differential voltage V_diff_, collected between points B and D, is a measure of the axial stress σ produced by the force F and, therefore, of the displacement dx.(12)$$V_{diff}=\xi (dx)$$

In the initial position, adjustable by screw 8 in Figure 5, the AEs are pre-stressed so that the working point is brought into the linear area of the characteristic in Figure 4d. With the modification of the trunk diameter by the value dx, the working point of the AEs moves on the characteristic producing a variation ΔZ_σ_ of the impedance AE, while the PEs remain unstressed having a constant value Z_p_.

An important disturbing factor is the temperature, which modifies the impedance with significant values, comparable to the variations due to the diameter change. The compensation of the temperature effect upon the device performance can be performed by mounting in a bridge the four identical elements, two active and two passive, similar to the strain gages approach. To obtain the dependence of the differential voltage of the bridge, V_diff_, on the impedance variation, ΔZ_σ_, one assumes that all four sensitive elements are identical in size, structure and properties. Applying a supplementary stress σ_bp_ to AEs for bringing the bias point to M, a supplementary impedance is added to the AEs, Z_bp_. Thus, the impedances of the four elements can be considered, in hypothesis, equal in the unstressed position, for a reference temperature θ_0_. Let us denote this impedance $Z_{\theta_{0}}$. For a stress σ and a temperature θ, we can write the following:(13)$${Z_{a}}_{\sigma \theta }={Z_{\theta_{0}}+Z}_{bp_{\theta_{0}}}+\frac{\partial Z}{\partial \sigma }{|}_{M}\Delta \sigma +\frac{\partial Z}{\partial \theta }{|}_{M}\Delta \theta$$(14)$${Z_{p}}_{\theta }=Z_{\theta_{0}}+\frac{\partial Z}{\partial \theta }{|}_{M}\Delta \theta$$
where

${Z_{a}}_{\sigma \theta }\;$is the impdeance of AEs at strain σ and temperature θ;

$Z_{\theta_{0}}$ is the impedance of unstressed wire at temperature θ_0_;

$Z_{bp_{\theta_{0}}}$ is the the bias point adjusting impedance of AEs at temperature θ_0_;

$\Delta Z_{\sigma }=\frac{\partial Z}{\partial \sigma }\Delta \sigma =S_{\sigma }\Delta \sigma$ is the variation in the impedance of the AEs due to the stress σ only, and $S_{\sigma }$ is the AE sensitivity to axial stress (viz. the slope of the characteristic in Figure 4d, S_σ_ = 12.4 mΩ/MPa;

$\Delta Z_{\theta }=\frac{\partial Z}{\partial \theta }=S_{\theta }\Delta \theta$ is the variation in the impedance of the AEs, as well as of PEs due to temperature. It includes also the variation $\Delta Z_{bp_{\theta }}$. S_θ_ is the sensor sensitivity to temperature, S_θ_ = 0.23%/°C. It is considered constant throughout the temperature domain;

${Z_{p}}_{\theta }$ is the impedance of PE at temperature θ, unstressed.

At a given temperature θ and at a given stress σ, the differential voltage of the sensor bridge, V_diffs_ becomes the following:(15)$$V_{diff_{s}}=\frac{I}{2}({Z_{a}}_{\sigma \theta }-{Z_{p}}_{\theta })=\frac{I}{2}(Z_{bp_{\theta_{0}}}+\Delta Z_{\sigma })$$
where I is the amplitude of the ac current that supplies the bridge.

It may be noted from (15) that the linearity of the SE characteristic is preserved, since V_diffs_ is proportional to ΔZ_σ_, but also that the temperature-dependent terms are eliminated. It should also be mentioned that any other influence that affects the four elements to the same extent, such as external magnetic fields, even terrestrial ones, will be eliminated by the same procedure.

By summarizing the above dependencies, we can conclude the following chain of conversions, according to Equations (1), (9)–(11) and (15):(16)d dx=mdd dx  F=kdx  F  σ=FSMAW  σ  ΔZσ=Sbpσ−σbp  ΔZσ  Vdiffs=I2Zbpθ0+ΔZσ Vdiffs  
where, referring to Figure 1:

d is the variation in the tree radius;

dx is displacement of the mobile element;

F is the additional force caused by the displacement dx;

σ is the stress that occurred in the MAW due to force F;

ΔZσ is the impedance variation due to stress σ;

$V_{diff_{s}}$ is the differential voltage that outputs the bridge.

Therefore, for a variation d in the trunk radius, a variation dx in the mobile element is produced, which, in turn, through the springs, acts with the axial force F on the SEs, which determines the stress σ and further, through the GSI effect, the variation in impedance ΔZ_σ_, that finally induces the voltage V_diffs_ in the measuring diagonal of the Wheatstone bridge. This entire chain is synthesized in (16).

### 4.3. Electronic Readout Circuit

To obtain the GSI effect, the sensitive element and, implicitly, the sensor bridge must be supplied with an ac current of frequency f > 100 kHz. As shown in Section 4.2, a trade-off among sensitivity, interference and signal-to-noise ratio can be carried out if the SEs are powered with a current of frequency of around 1 MHz and intensity of around 1 mA. The signal conditioning circuit diagram is shown in Figure 7.

This circuit consists of a Wien bridge oscillator that generates quasi-sinusoidal oscillations with a frequency of about 1 MHz and stabilized amplitude of about 2 V. The oscillator is followed by the voltage-to-current converter, whose constant current I alternately supplies two bridges: the sensor bridge (SB) containing SE and the reference bridge (RB), consisting of precise temperature-stable resistors, whose values are fixed to approximately the SE impedances. Why this solution was chosen is shown below. The two bridges are switched with the electronic switch S controlled by the microprocessor. The differential output voltage from the bridge is amplified, and then transformed into dc voltage using a peak-to-peak detector, whose output value will be the dc voltage:(17)$$V_{pp}=GV_{diff}$$
where G is the amplifier gain.

V_pp_ is further directed to a low-pass filter whose output, V_out_, is next converted into a digital word using the analog-to-digital converter ADC belonging to an ESP32 developing platform that, in turn, computes the output digital word N according to (21). N is finally transmitted wirelessly to a server for storage.

Throughout this chain, a problem is the inconsistency of the G gain due to time and thermal drifts of the amplifier, as well as the possible instability of the current source, which represent important sources of errors. The elimination of this inconvenience was performed using a ratio method based on the alternative measurement of the two bridges in the above scheme.

RB is composed of resistors R_p_, whose value is approximately equal to the value of the impedance Z_p_ at the working frequency and resistors R_a_, of a value corresponding to the impedance Z_aσ_, mounted on opposite arms of the bridge. RB is powered by the same current I as SB. R_p_ and R_a_ should be of good precision and stable with temperature. For this application, metallic foil resistors were employed having 1% tolerance and 10 ppm/°C temperature coefficient. Thus, RB outputs the following differential voltage:(18)$$V_{diff_{r}}=\frac{I}{2}G\left(R_{a}-R_{p}\right)$$
whereas, according to (15), the output of the sensor bridge is given by the following:(19)$$V_{diff_{s}}=\frac{I}{2}G\left(Z_{bp_{\theta_{0}}}+\Delta Z_{\sigma }\right)$$

The two voltages are fed to the ESP32 by means of the 12 bits A/D converter. The microcontroller computes the output value N as follows:(20)$$N=\frac{V_{diff_{s}}}{V_{diff_{r}}}=\frac{Z_{bp_{\theta_{0}}}+\Delta Z_{\sigma }}{R_{a}-R_{p}}$$
or, to summarize (16) in a final formula,(21)$$N=\frac{Z_{bp_{\theta_{0}}}+S_{bp}\frac{km_{d}d}{S_{MAW}}}{R_{a}-R_{p}}$$

The direct dependence of the output value of the device after signal processing, N, on the trunk variation, d, which can be used also as a theoretical calibration relationship can be seen from (21).

Since the switching and measurement of the two voltages is performed in a very short time interval, we can consider the values of I and G to be the same in both cases and, consequently, they are simplified within the ratio, obtaining for N an expression proportionally dependent only on ΔZ_σ_. In this way, the influence of the instability of the current source and of the gain G both in time and with temperature is eliminated.

## 5. Results and Discussion

The dendrometer was built as an experimental model and tested first in the laboratory under standard environmental conditions (constant temperature, pressure and humidity) to evaluate its performances. The characteristic of the transducer was thus plotted, that is, the dependence of the output number N against the displacement dx for three different temperatures:(22)$$N=\zeta (dx){|}_{\theta }$$

### 5.1. Experimental Setup

The characteristic was plotted for different positions of the bias point M using an experimental setup whose schematic view is shown in Figure 8.

The experimental setup contains a micrometric screw with a resolution of 0.05 mm to whose end the mobile part of the transducer is attached, so that its displacement dx is precisely measured. The characteristics were plotted over a range between 0 and 5 mm, the displacement being measured in 0.5 mm steps with the micrometer accuracy of 0.5%.

To evaluate the linearity and measurement range of the dendrometer, several characteristics were plotted with the bias point (zero) in various positions of the point M on the characteristic in Figure 4d, adjustable using the pre-stressing screw 8 in Figure 5.

To verify that the compensation scheme is effective, the characteristics were plotted also for the following three different temperatures: 5 °C, 25 °C and 45 °C. For this, the transducer was placed in a thermostatic enclosure model Steinberg SBS-LI-18, with the electronic part remaining outside the enclosure. The voltage collected from the sensor was brought to the circuit via a twisted cable of maximum 0.5 m length. Each set of measurements was performed after the temperature had stabilized within ±1 °C. Since the transducer bridge was powered with constant current, no shielding measures were required for the power cable.

The error calculation was carried out using the classical method described in the Guide to the Expression of Uncertainty in Measurement (GUM). In this case, the uncertainties have two components: the random component (type A uncertainties), resulting from various influences such as non-uniformities in the internal structure of the wires, mechanical imperfections and deviations and variations in the spring constant, and type B uncertainties, derived from the systematic error of the micrometer, which is 0.5%, according to its technical datasheet.

To assess the measurement uncertainty, 12 sets of measurements were performed for each point of the characteristic curve. For this purpose, 6 round-trip displacements of the moving element were carried out over the 0–5 mm–0 range, with a 0.5 mm step. For the values obtained at each point, the mean value and the standard deviation were calculated, along with the confidence intervals for each point at a 95.5% confidence level, using the Student’s distribution.

The relative error was determined by normalizing the confidence interval bounds to the mean value of each point. The widest confidence interval among all points was then quadratically averaged with the micrometer’s error. Repeatability was calculated similarly, but by considering the relative variation in σ with respect to the mean value at a given point.

All measured values of N were sent by the microcontroller to the cloud and recorded in a Google Docs sheet as decimal values.

### 5.2. Characteristics

In Figure 9a, the characteristic of the transducer N = ζ(dx) is plotted for the temperature of 25 °C and the bias point M corresponding to an axial pre-stress of approximately 180 MPa. In Figure 9b,c, the characteristics plotted for axial pre-stress of 300 MPa and without pre-stressing are given. The strong nonlinearity of the characteristics observed in cases (b) and (c) is due to the nonlinearity of the sensitive elements, but also to the sensor bridge in which they are mounted. For variant (a), the best linearity of the device was 1.4%. For the most linear characteristic, corresponding to the bias point for σ = 180 MPa, the calculated uncertainty computed as explained in the previous section was 5.3% for a confidence interval of 95.5% and the measurement range was 0 to 5 mm, whereas the repeatability resulted in 2.8% at the temperature of 25 °C.

In Table 1, the values of N corresponding to all three temperatures are given. The temperature coefficient K_θ_ was computed as follows:(23)$$K_{\theta }=\mathrm{ma}x_{i}\left(\frac{{\epsilon_{µ}}_{i}(\theta )}{\Delta \theta }\right)=\mathrm{ma}x_{i}\left(\frac{{\left(\frac{\Delta {µ}_{i}}{{µ}_{i}}\right)}_{\theta }*100}{\Delta \theta }\right)\;$$
where ${\epsilon_{\mu }}_{i}\left(\theta \right)$ is the relative variation in the mean due to the temperature variation for point i.

The average value of the temperature coefficient obtained following the experiment is K_θ_ = 0.012%/°C. This value is much lower than the temperature coefficient of the MAW permeability, which is between 0.07 and 0.125%/°C, depending on the permeability [26].

### 5.3. Error Sources and Performances

The measurement with the device proposed in this paper is disturbed by several sources resulting in errors that disrupt the measurement accuracy. This accuracy was experimentally evaluated under laboratory conditions as a confidence interval ±2σ (σ is the standard deviation). According to the plotted characteristics, the assessed uncertainty was 5.3%.

Considering that the device is designed to work in the outdoor environment, it will be affected by significant temperature variations. According to the results in Table 1, the effect of temperature on the device is reduced; however, it is not negligible. For in-land operation, temperature contributes not only to the modification of the properties of the sensitive elements, but also to dimensional changes in the device due to the thermal coefficients of the strip and of the plastic material from which the parts are made. In such conditions, a global assessment, on land, of the influence of temperature on the final result is necessary and a supplementary compensation must be achieved by assisting the measurements with a temperature sensor and correcting the characteristics by software.

To the same extent, the influence of external magnetic fields upon the sensor may be consistent. However, these influences are compensated similarly with temperature by using the complete bridge of elements.

Another source of errors would be the bark microrelief found on the trunk surface, especially when these bumps are large and would affect the adherence of the strip to the trunk. A solution to compensate for this drawback is described in [27].

Special attention must be also paid to the tolerance of the fixed resistances of the reference bridge. Since in (21), at the denominator a difference exists, the effect of the resistance tolerance upon the computed quantity N becomes significant. A Monte Carlo analysis on (21) considering the resistances R_a_ and R_p_ having a tolerance of 1% leads to a maximum error in the results of 4.2%. It is therefore necessary to select these resistors or use better precision devices; however, this significantly increases the cost of the final product. Also, even if temperature-stable metal foil resistors are employed in the reference bridge (K_θ_ resistor = 10 ppm/°C), the experimental results reveal a greater influence of temperature on the overall device (K_θ_ device = 0.012%/°C) due to the same reason.

Another source of error, which actually provides the dendrometer resolution, is the resolution of the A/D converter. The processing circuit uses the built-in ESP32 12 bit A/D converter for the acquisition and conversion of the output voltages from the two bridges into digital words. The conversion quantum corresponding to an LSB is 0.8 mV, this represents a variation in the computed N of 0.36% of the corresponding bias point position, which is an error introduced by the quantization noise. This leads to a minimum resolution of the entire device of 1.6 µm.

Other sources of errors may be mechanical construction imperfections, variations in the spring constant with temperature, dispersions of the SE characteristics due to the manufacturing process or improper mounting of the device on the tree trunk.

Finally, the metrological performances obtained with the experimental device, under laboratory conditions, are summarized in Table 2.

The tests conducted in the laboratory on the proposed device showed a performance comparable to those of devices reported in the literature [11,12,13,14,15,16] and available on the market (e.g., ICT International, Australia, Dynamax, Fallstone, Houston, TX, USA, Tomst, Praha, Czech Republic, Ecomatik, Dachau/Munich, Germany). The device presented in this paper is currently at the experimental prototype level, but its performance can be significantly improved in the case of a technology transfer.

For example, the resolution of commercial dendrometers ranges between 1 µm and 4 µm, while our device achieves a resolution of 1.4 µm after signal processing, with the potential for further improvement by using a higher resolution A/D converter (e.g., 14 or 16 bits). Although the precision reported in the literature ranges between 2% and 5%, the current precision of our dendrometer is 5.3. It can be greatly improved; however, through a more precise mechanical design and by using high-quality wires with lower characteristic dispersion.

The functionalities of the device can be extended by adding new sensors to monitor parameters such as temperature, humidity, precipitation or electrodes for monitoring sap flow in the trunk. These combined measurements can provide much more comprehensive information about the plant’s biological activity.

Furthermore, the performance can be enhanced by supplementing the signal processing chain at the microcontroller level with various filtering, linearization and systematic error compensation operations.

Last but not least, the production cost of our device can result significantly lower than that of commercial devices (which range between USD 500 and USD 1000), considering that the sensitive element—the amorphous wire—costs approximately USD 1.5 per meter.

### 5.4. In-Field Deployment

To assess the behavior of the device in the natural environment, the dendrometer was tested in-field by attaching it to a tree of approximately 35 cm diameter, as shown in Figure 10.

The experiment was performed for a period of 27 days in August 2024. The dendrometer was attached to an apple tree in an orchard located in a temperate continental region in Iaşi, Romania. The chosen period was characterized by variations in temperature from a minimum of 14 °C during some nights to a maximum of 36 °C during some days. There was sporadic rain on some days. The precipitation volume was not recorded. The tree was placed in the area of a Wi-Fi router, so that the information was sent to the cloud sheet, just like in the laboratory testing.

Figure 11 shows the graph of trunk variation, d, measured with the proposed dendrometer over a period of 27 days, between August 1st and August 27th, 2024. This graph was plotted in order to qualitatively verify the functionality of the device, and not to evaluate the sensor’s resistance to various aggressive environmental factors, which may occur under adverse weather conditions or during the winter, nor to make any interpretation regarding the biological behavior of the tree during this period.

In Figure 12, the detail of 6 days from the 27 days of the recorded period is extracted, in which the trunk extensions and contractions due to the circadian cycle are traced, along with the temperature recording in the same period. The dendrometer was not previously calibrated for the diameter of the tree, so the values obtained are not accurate. It is observed from the figure that as expected, the tree trunk shrinks at high temperatures during the day because of the lower hydric status, whereas during the night the trunk recovers its dimension. From recordings over the whole period, slighter variations were observed during the rainy days. However, no conclusions can be drawn from Figure 11 and Figure 12 regarding the overall tree’s hydric status or health condition, as longer-term recordings are required for such assessments. In this regard, a project has been initiated in partnership with the University of Life Sciences in Iași, Romania, for long-term monitoring (over a period of at least one year), with interpretation of the results by specialists in horticulture from this university, and simultaneous evaluation of the dendrometer’s robustness and endurance.

The experiment was conducted just for observing the device behavior when working in the field, without dealing with the measurement accuracy.

## 6. Conclusions

This paper describes the principle of operation, construction and experimental results obtained from testing a device called a dendrometer, which is designed for measuring small variations in tree trunks. As its sensitive elements, the dendrometer employs magnetic amorphous wires which under the action of the stress impedance effect, converts with high sensitivity the small movements of a mobile element caused by the variations in the trunk diameter into axial stress, then into impedance variations and, finally, into an electrical signal proportional to the diameter variations. Under certain design and optimization conditions, satisfactory or superior metrological performances are obtained compared to dendrometers existing on the market, while having at the same time the advantage of better sensitivity and resolution, immunity to temperature variations, the possibility of measuring regardless of the shape of the trunk section and, last but not least, a low cost.

As for the next stages of development, ongoing research is being addressed to study the device’s long-term behavior in the laboratory and in-field, draw conclusions regarding the influence of time on its metrological properties and performance and collect more data to analyze and interpret using artificial intelligence.

## Figures and Tables

**Figure 1 sensors-25-04449-f001:**
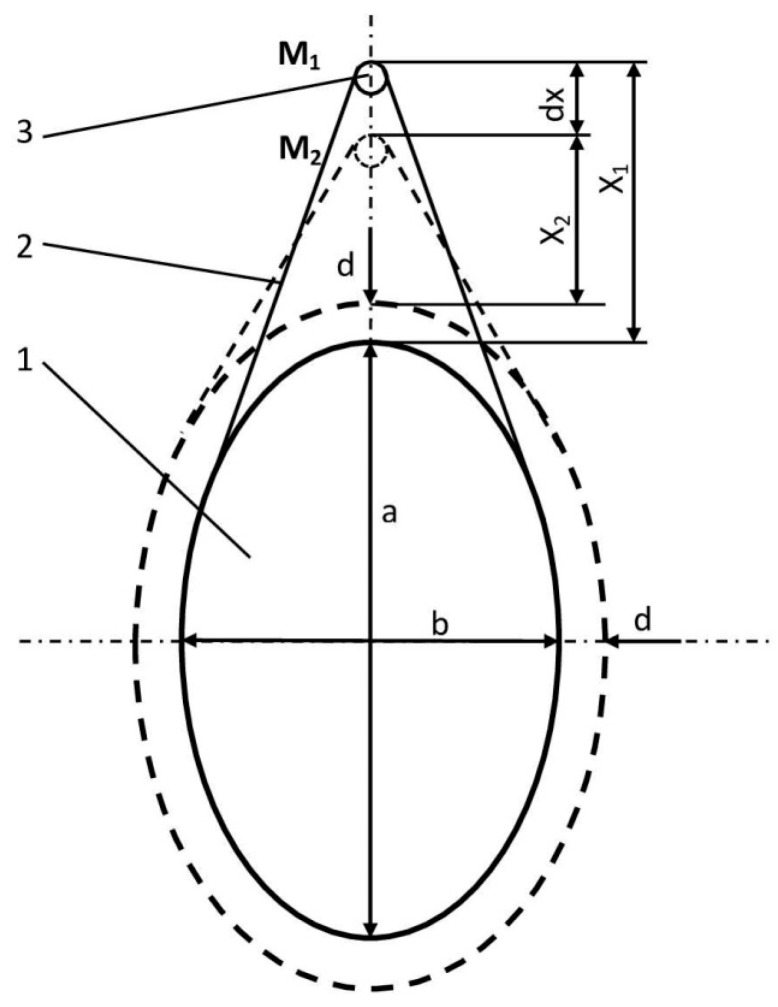
Illustration of the operation principle of the dendrometer: 1—tree trunk; 2—flexible strip; 3—mobile point.

**Figure 2 sensors-25-04449-f002:**
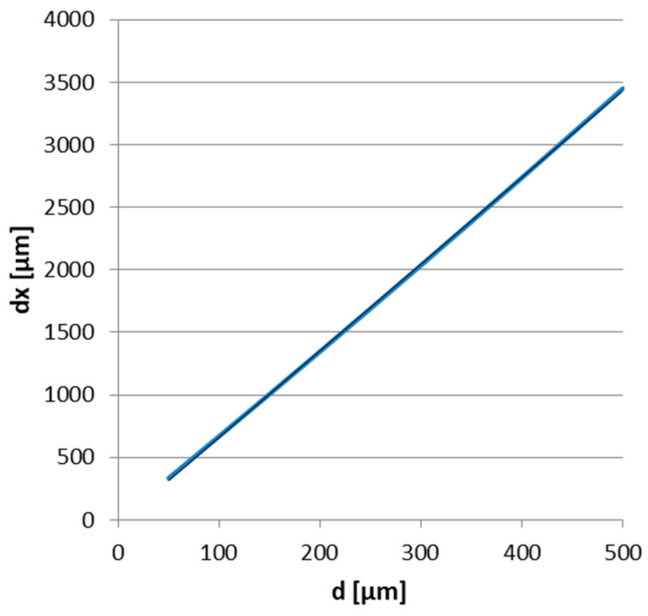
Example of dependence of dx on d for X_1_ = 20 mm, a = 155 mm and b = 140 mm.

**Figure 3 sensors-25-04449-f003:**
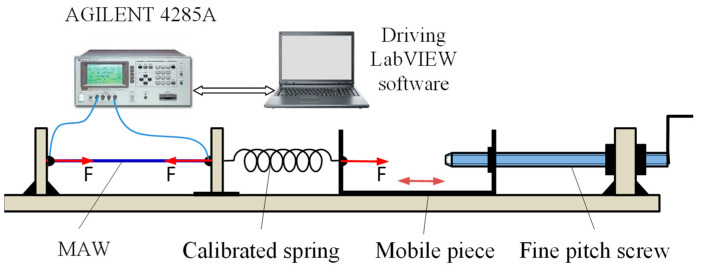
Experimental setup for tracing the characteristics of the sensitive elements.

**Figure 4 sensors-25-04449-f004:**
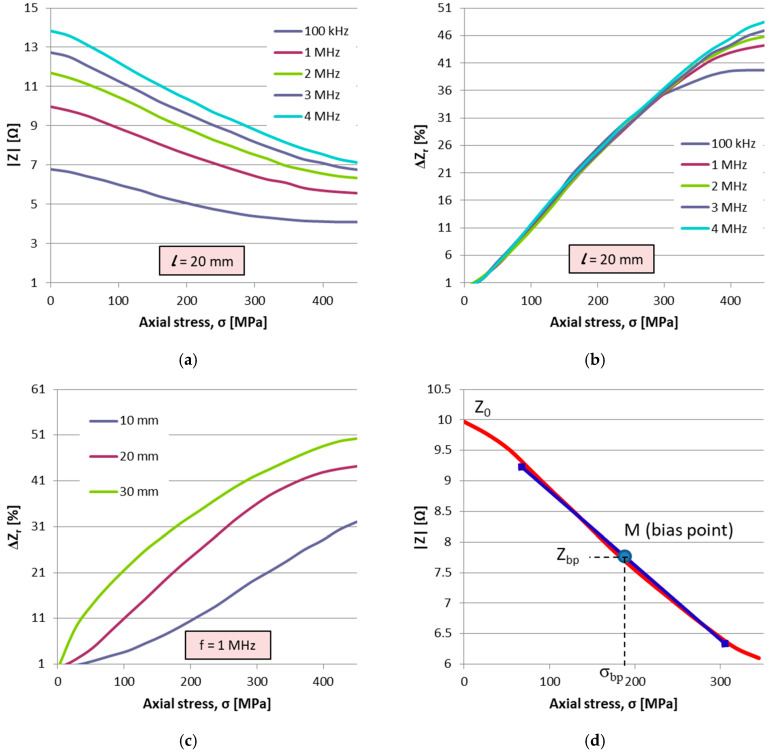
(**a**,**b**) Variation in the impedance and sensitivity for l = 20 mm, I = 1 mA and various frequencies, (**c**) variation in sensitivity for various lengths, f = 1 MHz and I = 1 mA and (**d**) optimized characteristic for l = 20 mm, f = 1 MHz and I = 1 mA.

**Figure 5 sensors-25-04449-f005:**
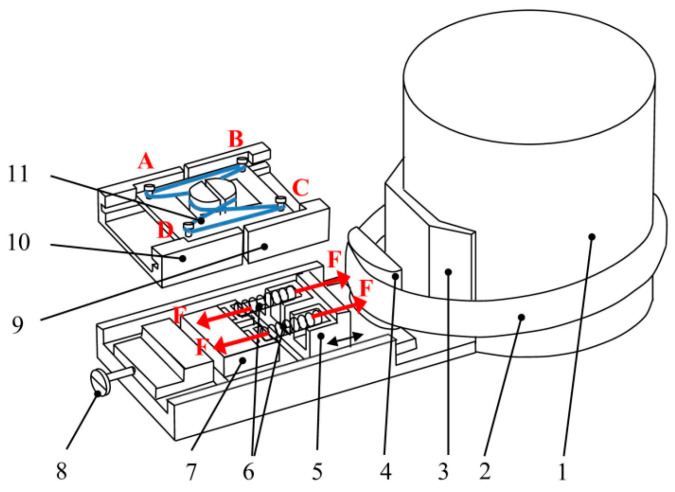
The mechanical structure of the dendrometer. 1—trunk of the tree, 2—stainless steel metallic strip, 3—fixed casing, 4—shoulder, 5—movable piece, 6—calibrated springs, 7—fixed piece linked to the casing 3, 8—pre-stressing screw, 9, 10—parts linked to 5 and 7, 11—magnetic amorphous wire.

**Figure 6 sensors-25-04449-f006:**
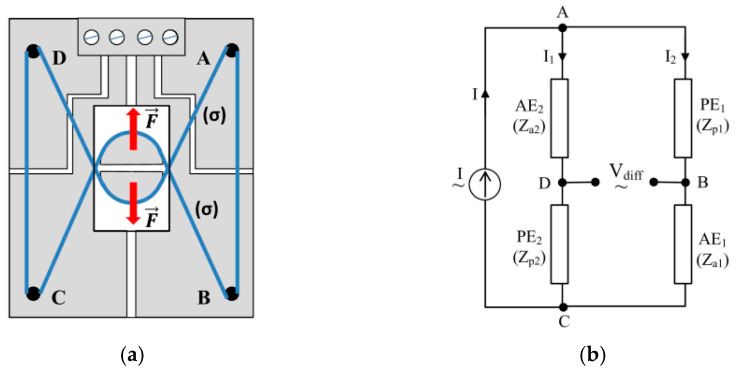
(**a**) Wheatstone bridge composed of two active sensitive elements (AE_1_ and AE_2_) and two passive elements (PE_1_ and PE_2_) and (**b**) the electrical circuit.

**Figure 7 sensors-25-04449-f007:**
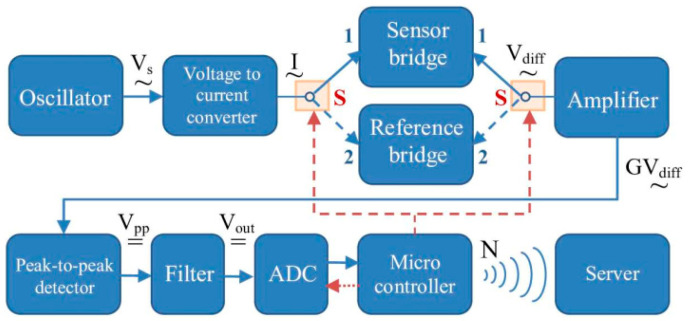
Block diagram of the signal conditioning circuit.

**Figure 8 sensors-25-04449-f008:**
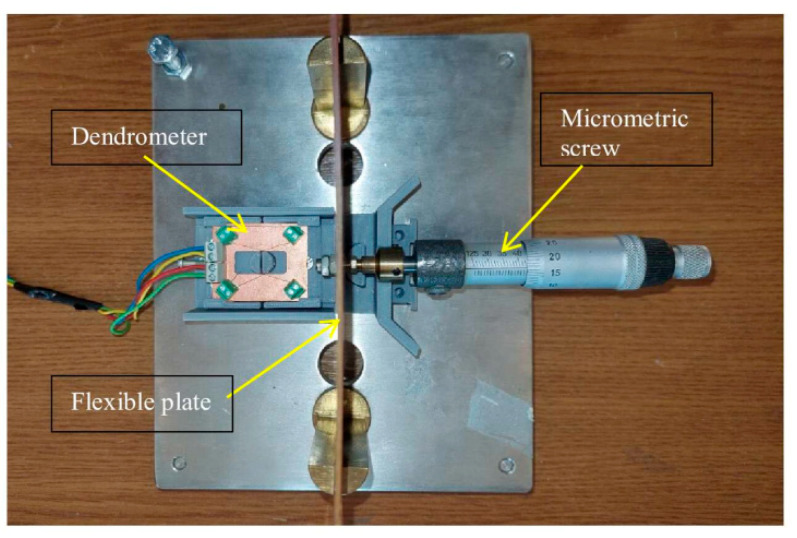
A view of the experimental setup for device testing.

**Figure 9 sensors-25-04449-f009:**
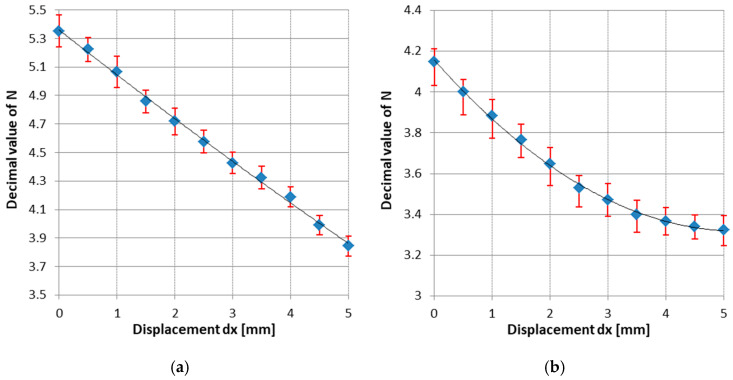
The characteristics of the dendrometer for θ = 25 °C and (**a**) σ_bp_ = 180 MPa, (**b**) σ_bp_ = 300 MPa, (**c**) σ_bp_ = 0 MPa.

**Figure 10 sensors-25-04449-f010:**
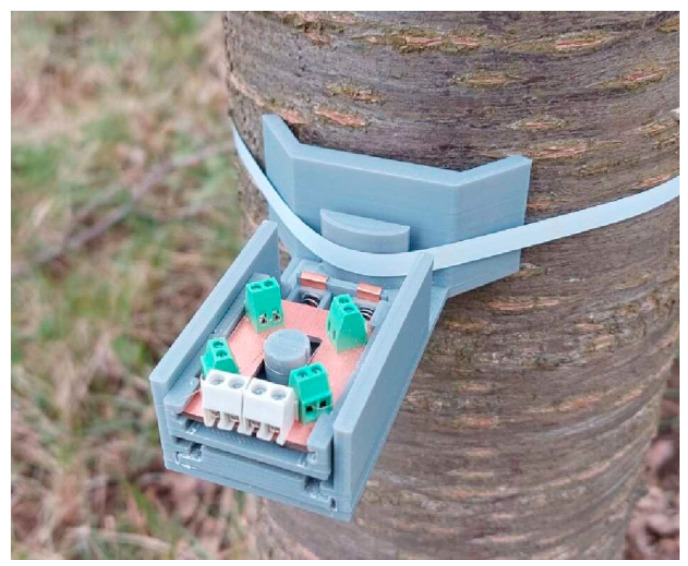
The dendrometer attached to the trunk.

**Figure 11 sensors-25-04449-f011:**
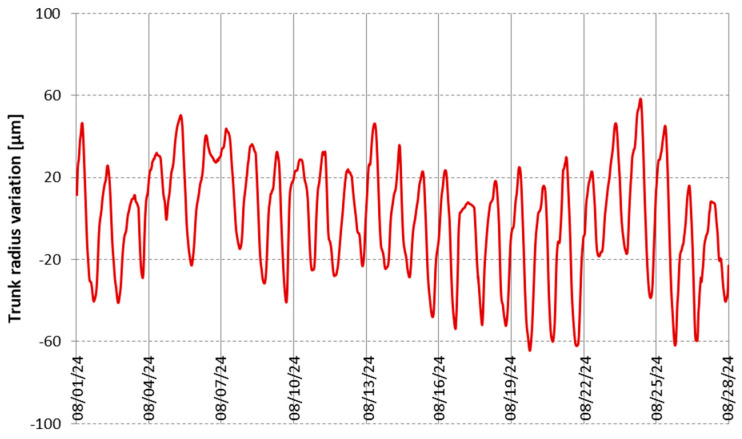
Recording during 27 days in August 2024 of trunk radius variations.

**Figure 12 sensors-25-04449-f012:**
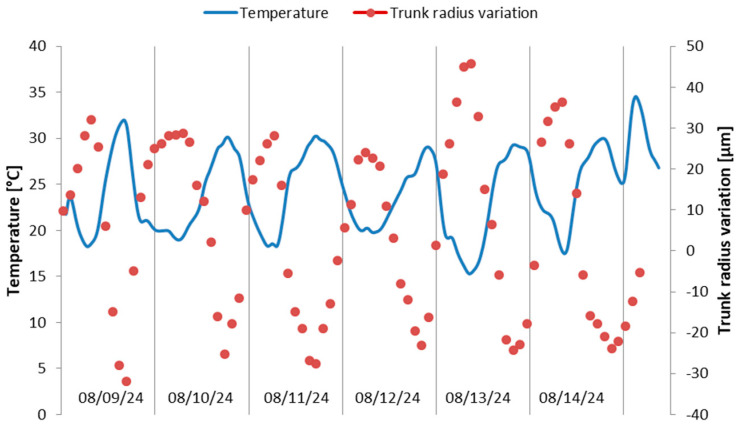
Recording during 6 days in August 2024 of temperature and trunk radius variations.

**Table 1 sensors-25-04449-t001:** Variation in the sensor output with temperature.

θ [°C]	dx [mm]	−2	−1	0	1	2
5	N	5.162	4.872	4.576	4.281	3.898
25	N	5.176	4.882	4.588	4.294	4.000
45	N	5.192	4.890	4.600	4.305	4.012

**Table 2 sensors-25-04449-t002:** Summary of the metrological parameters.

Parameter	Value
Measurement range	0–5 mm
Linearity	1.4%
Accuracy	5.3% of full scale
Resolution	1.6 × 10^−6^ m
Repeatability	2.8%
Temperature coefficient	0.012%/°C
Power demand in active state	Less than 100 mW
Response time	Less than 1 s

## Data Availability

The original contributions presented in this study are included in the article. Further inquiries can be directed to the corresponding author.

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
