# Peer review of "A New Device for Measuring Trunk Diameter Variations Using Magnetic Amorphous Wires"

_sensors, 2025, doi:10.3390/s25144449_

Round 1
Reviewer 1 Report
Comments and Suggestions for Authors
Thank you for submitting your research paper entitled "A New Device for Measuring Trunk Diameter Variations Using Magnetic Amorphous Wires". This study innovatively leverages the Giant Stress Impedance effect in magnetic amorphous wires to develop a novel apparatus for monitoring trunk diameter variations. The implementation of a full-bridge temperature compensation scheme and optimized parameters for the sensitive element significantly enhances the device’s measurement sensitivity and environmental robustness.
This MAW-based sensing approach establishes a novel pathway for precision agriculture and forest ecosystem monitoring. We look forward to seeing test results of the device across a broader range of tree species and environmental conditions.
I think the proposed idea is very attractive, but there are a few small suggestions. Please see the attachment.

Reviewer 2 Report
Comments and Suggestions for Authors
See attachment

Reviewer 3 Report
Comments and Suggestions for Authors
The authors presented a novel dendrometer utilizing magnetic amorphous wires to detect minimal variations in tree trunk diameter with high sensitivity and resolution. The device's design incorporates a mechanical structure made from 3D-printed parts and an electrical bridge circuit that converts axial stress on the amorphous wire into an electrical signal, facilitating precise measurements. Unlike traditional dendrometers that rely on displacement transducers, this approach offers immunity to temperature effects, shape irregularities, and environmental noise, enhancing measurement reliability. Laboratory and field tests demonstrate the device's ability to monitor diurnal patterns of trunk expansion and contraction related to temperature and water status, albeit with some calibration limitations. The system’s low cost and potential for wireless data transmission make it suitable for long-term forest monitoring applications. However, challenges remain in calibrating the device for diverse tree species with irregular trunk shapes and in ensuring long-term stability in harsh outdoor conditions. Overall, this research advances dendrometry technology by offering a sensitive, cost-effective, and adaptable solution for monitoring tree dynamics. Future work will need to focus on refining calibration procedures and assessing long-term performance in varied ecological settings.
Several points need to be clarified:
1 Not all experimental dependencies have error bars. It is necessary to describe the error assessment in more detail. What was the repeatability of the measurements? Are the graphs for a single measurement provided in the work?
- The authors claim that the measurements were taken over 27 days, but they only give an example of a daily graph of tree size fluctuations lasting several days. It makes sense to provide data for all 27 days of continuous measurements.
- The authors do not separate the effect of temperature on wood and on the compression of the sensor wire. How did the authors take into account the thermal expansion coefficient of the wire? It is necessary to make appropriate thinnings.
After these minor changes, the article can be published.
Round 2
Reviewer 1 Report
Comments and Suggestions for Authors
Thank you for your patient and thorough responses to the four issues I raised. Your answers addressed each point accurately and comprehensively, and the revisions to the paper are now exceptionally polished. I have no further suggestions at this stage.
Reviewer 2 Report
Comments and Suggestions for Authors
publish as is